# Linear-to-circular polarization conversion with full-silica meta-optics to reduce non-linear effects in high-energy lasers

Nicolas Bonod [1] ✉, Pierre Brianceau[2], Jérôme Daurios[3], Sylvain Grosjean [3], Nadja Roquin[3], Jean-Francois Gleyze [3], Laurent Lamaignère [3] ✉ & Jérôme Neauport [3] ✉

High-energy lasers have benefited from intense efforts to bring light-matter interactions to new standards and to achieve laser fusion ignition. One of the main issues to further increasing laser energy is the resistance of optical materials to high laser fluences, in particular at the final stage of the laser beamline where nonlinear Kerr effects can occur in optical materials and provoke laser filamentation. One promising way to mitigate this process is to reduce the nonlinear susceptibility of the material by switching the polarization from a linear to a circular state. Here, we report a significant reduction in the laser filamentation effect on glass by using a full-silica metamaterial waveplateable to switch the linear-to-circular polarization of high fluence laser beams. This result is achieved through the use of a large size full-silica meta-optics exhibiting nominal polarization conversion associated with an excellent transmission efficiency and wavefront quality, as well as a high laser damage resistance.

Since the invention of the laser in 1960[1], high energy, high peak power laser facilities have seen decades of developments driven by multiple applications ranging from particle acceleration, high density and high pressure physics to astrophysics and imaging. The quest for fusion energy has motivated the development of dedicated high energy laser facilities operating in the UV spectrum such as the National Ignition Facility (NIF, United States) and Laser MegaJoule (LMJ, France). Dozens of high power laser facilities are currently in operation spanning worldwide[2,3] with, for some of the most powerful or energetic, chirped pulsed amplification based facilities overcoming the 10 PW barrier in the infrared such as ELI-NP[4] or neodymium phosphate flash-lamp pumped frequency tripled energetic laser system such as the NIF exceeding the 2 MJ level of ultraviolet energy[5,6]. The pursuit of the path to higher increased energy and peak-power has seen over last months major outcomes and announcements. Among these, the recent results of the NIF achieving a record fusion energy yield of more than 1.3 MJ[7–9] and more recently the first-ever controlled fusion reaction yielding an

energy gain[10] have stimulated interest in energy production with high energy laser facilities[11–13]. The up-scale of high-energy laser facilities is currently limited by several challenges. In addition to bandwidth constraints and aperture size limitations induced by optics manufacturing techniques, one of the main challenges is to increase laser damage resistance of optical materials in operating conditions. Several efforts are being made to improve the performance of laser optics manufacturing processes such as fused silica finishing process improvements[14,15], multi-layer dielectric coatings optimization[15,16], nanostructures engineering[17–23]. More disruptive concepts such as replacing optical materials by plasmas[24,25] were also proposed but still need to be developed and implemented. Another major challenge relies on mastering non-linear effects such as Raman and Kerr effects[3,26]. The latter is characterized by the B-integral given by the relation:

$$B = \frac{2\pi}{\lambda} \int_0^L n_2(x,y,z) I(x,y,z,t)\, dz, \qquad (1)$$

[1]Aix-Marseille Univ, CNRS, Centrale Marseille, Institut Fresnel, Marseille 13013, France. [2]CEA GRENOBLE, Grenoble F-38054, France. [3]CEA CESTA, Le Barp F-33116, France. ✉e-mail: nicolas.bonod@fresnel.fr; laurent.lamaignere@cea.fr; jerome.neauport@cea.fr

where $\lambda$ is the wavelength, $n_2$ the nonlinear refractive index of optical materials, $I$ the intensity of the laser beam, $z$ the direction of propagation and $L$ the cumulative length of optical materials along the beamline. Intense efforts have been made to reduce this B-integral; Namely, (i) using optical materials featuring small nonlinear susceptibilities, in particular fused silica optics in transmission[27] and low $n_2$ amplifier medium[28], (ii) improving beam contrast by the use of spatial light modulators in front-end for global[6,29–31] and local[5,32] corrections, (iii) understanding and limiting FM-to-AM (Frequency Modulation-to-Amplitude Modulation) conversion in laser beam lines to reduce temporal modulations[33–37]. The combination of the various conditions leading to the formation of filaments has been the topic of several contributions[38,39] and efforts are still underway to better control and push back the break-up threshold mainly through a better understanding of the impact of amplitude and phase perturbations[40,41].

Here we propose to exploit the dependence of the non-linear properties of materials on polarization to further reduce the B-integral in silica by changing the state of polarization of the incident beam. For transparent materials in the nanosecond regime, the nonlinear index of refraction is mostly induced by the atomic nonlinear polarizability which includes nuclear and electronic contributions[42]. In isotropic materials such as fused silica, by neglecting the nuclear contributions, the non-linear refractive index in linear $n_2^{linear}$ and circular polarization $n_2^{circular}$ can be written as (see Eqs. 11b,12b in ref. [42]):

$$n_2^{linear} = \frac{12\pi}{2n_0}\chi_{1111}^{(3)} = \frac{3\pi}{2n_0}\sigma, n_2^{circular} = \frac{24\pi}{n_0}\chi_{1122}^{(3)} = \frac{\pi}{n_0}\sigma, \qquad (2)$$

where $n_0$ is the linear index of refraction, $\chi^{(3)}$ is the third order susceptibility tensor and $\sigma$ a Born-Oppenheimer coefficient describing electronic contributions. These expressions show the dependency of the nonlinear refractive index $n_2$ on the polarization state but they also show that the nonlinear index of refraction in glass is reduced in circular polarization by a factor of 3/2 compared to the linear polarization. Direct $n_2$ time-resolved interferometry measurements at the wavelength of 1053 nm on SF6 glass and YAG crystal by Moran[43] confirmed this 3/2 proportionality term between circular and linear polarizations. Soileau et al. measured the same 3/2 factor on fused silica thanks to optical breakdown experiments carried out at $\lambda = 1053$ nm and $\lambda = 532$ nm[44,45].

While high-power laser systems are generally set-up to operate in linear polarization which is natural and easy to manipulate, switching to circular polarization in the final stages is a promising way to reduce the B-integral and therefore the resistance of optical materials to laser filamentation. However, if commercial wave-plates based on birefringent materials such as quartz are routinely used in optical systems, they cannot be implemented in high-energy laser chains because the large size of the optics is not compatible with the crystalline nature of quartz. Amorphous fused silica is a material of special interest to operate at such laser fluences. Amorphous silica is an isotropic material and the anisotropy must be engineered by nanostructuring the amorphous material. It has been shown in ref. [46] that amorphous silica etched with a pitch of 260 nm together with a high aspect ratio with groove depths typically around 750 nm behaves as a $\lambda/5$ birefringent waveplate.

Here, we evidence that nanostructured silica fabricated close to the nominal configurations can achieve a $\lambda/4.3$ value together with a high transmittance exceeding 95% (corresponding to 99% for the nanostructure alone) and low wavefront error down to the millimeter scale spatial period. The waveplate is illuminated in normal incidence with a linearly polarized beam at a wavelength of $\lambda = 351$ nm. The polarization state can be simply controlled by the orientation of the nanostructured waveplate, i.e., the orientation of the linearly polarized incident electric field with respect to the axis of the linear grooves on silica. The polarization state is not affected

by the waveplate when the incident linearly polarized electric field is parallel or orthogonal to the groove axis, while it is switched when the incident electric field is oriented at 45° from the groove axis. We report a drastic effect of the incoming beam polarization tailored by the metamaterial waveplate on the laser filamentation. Experiments carried out in the nanosecond regime at $\lambda = 351$ nm clearly show how switching from a linear to a circular polarization cancels the formation of filaments at a mean fluence of 5.7 J/cm² and drastically reduce the number of filaments at a higher mean fluence of 7 J/cm². The number of filaments found in circular polarization at 7 J/cm² is even smaller than that obtained in linear polarization at the fluence of 5.7 J/cm².

## Results
### Optical characterization of the silica quarter waveplate
The design of a metamaterial quarter waveplate is driven by the need to combine a $\lambda/4$ phase shift with a high transmittance, typically above 96%. Let us recall that the waveplate is illuminated in normal incidence with a laser beam at $\lambda = 351$ nm. It was showned in ref. [46] that the transmittance is mainly controlled by the period of the grooves while the phase shift is mainly driven by the depth of the grooves. A short period typically smaller than 260 nm is required to achieve a high transmittance and an important groove depth, typically >800 nm and ideally equal to 1000 nm, is required to achieve the targeted $\lambda/4$ phase shift.

From a manufacturing point of view, we aim to (i) achieve the nominal configuration in terms of phase shift and transmittance by targeting a period as short as 230 nm and a groove depth of 1000 nm, which corresponds to an aspect ratio of 4.35. It should be noted that this short period of 230 nm avoids the propagation of diffractive orders in normal incidence and that this waveplate reflects and transmits only the specular order. The targeted value of the Duty Cycle (DC) that is equal to the ratio between the wall width at mid-height over the period is 0.4. We also aim at (ii) fabricating samples by nanostructuring silica over a $110 \times 110$ mm² area centered on a 200 mm diameter fused silica wafer.

Finally, we aim to (iii) associate two orientations of the unixial nanostructured silica. For that purpose, the silica walls are oriented at −45° and +45° from the vertical $Oy$-axis on respectively the left and right part of the waveplate with a continuity of the walls between the two zones, as displayed in Fig. 1a, b). The nanostructured silica waveplate is fabricated through an electron beam lithography process (see "Methods"). The vertical and straight silica walls associated with a high aspect ratio are obtained by the use of a hard mask composed of a titanium layer. The analysis of images obtained by Scanning Electron Microcroscopy (SEM) displayed in Fig. 1b provides information on the depth ($h$), period ($d$) and duty cycle obtained with values estimated to be $h = 1000$ nm, $d = 230$ nm and DC = 0.4 respectively.

The spatially resolved linear retardance was measured at 355 nm in transmission in normal incidence (see "Methods"). A linear retardance of 82.75 nm ± 0.77 nm (2 standard deviations) and 82.20 nm ± 0.93 nm (2 standard deviations) are obtained over the clear aperture combining the two areas (Fig. 1c). This corresponds to a $\lambda/4.3$ waveplate, hence very close to the quarter waveplate targeted. This metaoptics behaves as a uniaxial crystal[47] and the difference between the ordinary and extraordinary index $\Delta n$ can be easily estimated from the measurement of the linear retardance and the nanostructuring depth at $\Delta n = 0.0825$.

Figure 1d presents the angle of the neutral axis of the waveplate. Thanks to the e-beam lithography technique, an almost perfect orientation of the fast axis is obtained with mean angles of +44.89° ± 0.02° and −44.89° ± 0.02° for the left and right areas respectively. These results demonstrate that the large scale nanostructure behaves as a quarter waveplate producing left and right

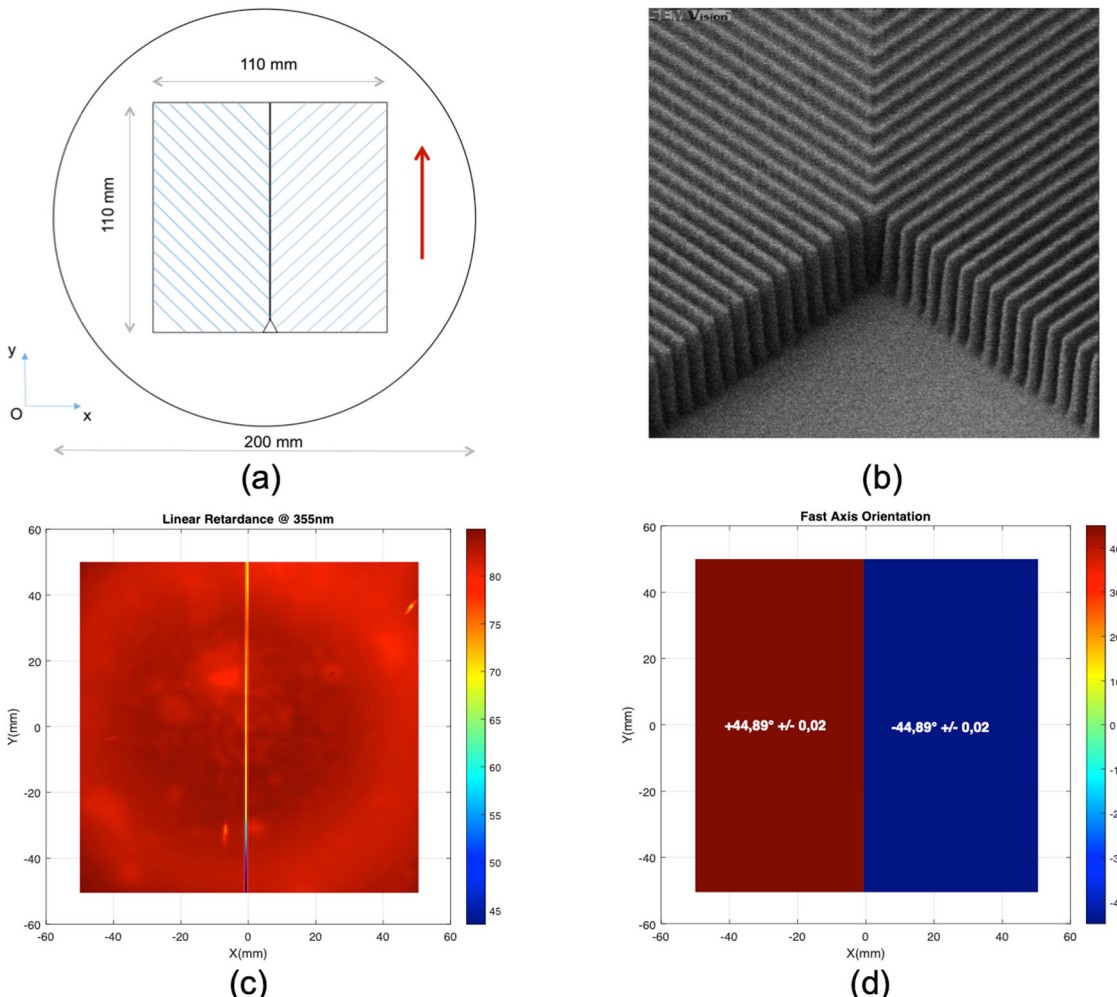

**Fig. 1 | Fabrication and linear retardance of the full-silica quarter waveplate.**
**a** Template of the quarter waveplate. Blue lines are guide to the eyes to represent the orientation of the grooves of the left and right areas oriented at 90° to each other. **b** SEM image of a central area of the waveplate. A perfect lining of the silica pillars between the two waveplate areas can be observed. Image width is 6 μm approximately while the full size of the sample is 110 × 110 mm² centered on a 200 mm diameter fused silica wafer. **c** Spatial distribution of the linear retardance
measured at λ = 355 nm at normal incidence on the full-scale transmission wave-plate. A mean linear retardance of ~82 nm is measured with an overall excellent homogeneity over the 110 × 110 mm² area of the waveplate. **d** Experimental angle of the fast axis of the waveplate measured at λ = 355 nm: a symmetric distribution of this angle relatively to the Oy-axis is observed with values of +44.89° ± 0.02° and −44.89° ± 0.02° on the left and right areas of the bizone waveplate.

circular polarizations for an incoming linear polarization oriented in a vertical direction, *i.e.* along the transition line separating the two zones of the waveplate (Fig. 1a).

The spatially resolved transmission efficiency of the waveplate was measured at 351 nm in normal incidence (see "Methods"). Figure 2a shows the spatial distribution of the transmission efficiency when the waveplate is illuminated in normal incidence with a linear polarization oriented along the Oy-axis (Fig. 1a). Mean values of 0.953 ± 0.0049 and 0.955 ± 0.0049 (2 standard deviations) are obtained for the left and right areas respectively (uncertainty at two standard deviations). These transmission efficiencies include the reflection from the uncoated fused silica back side of the waveplate which value is equal to 0.0367. For an incoming polarization oriented along the horizontal along the Ox-axis (Fig. 1a), mean values of 0.947 ± 0.0049 and 0.9495 ± 0.0049 are obtained for the left and right areas respectively as shown in Fig. 2b. Consequently, the nanos-tructure alone exhibits a mean diffraction efficiency of 0.98 or 0.99 depending on the polarization orientation with a high uniformity over the whole clear aperture.

Small discrepancies in terms of diffraction efficiency and transmitted wavefront (Fig. 2) observed when illuminating the

waveplate in horizontal or vertical directions are due to the slight deviation of the waveplate from a quarterwave. The missing retardance of 5 nm corresponds to a grating depth error of 50 nm[46]; it produces a weak elliptical polarization at the origin of the differences.

The challenge of the implemention of meta-optics in high energy laser facilities is to achieve wavefront aberrations of the same order as those obtained with conventional planar dioptric optical components[18]. Mid and high spatial frequencies of 0.1–10 mm⁻¹ can be detrimental for the operation and performances of high energy laser systems[15,48,49]. We therefore measured the transmitted wavefront quality of the nanostructured quarter waveplate in normal incidence (See the Methods section "Transmitted wavefront"). The distribution over the meta-optics of the spatially resolved transmitted wavefront displayed in Fig. 2c and d has a peak-to-peak amplitude of 18 and 20 nm for the vertical and horizontal orientation respectively and an RMS in the [1–10 mm] period range of 1.4 nm. Figure 2e represents the Power Spectral Density (PSD) calculated from the transmitted wavefront measurement displayed in Fig. 2d. The PSD does not exhibit frequency peak beyond the boundary (green line in Fig. 2e) usually retained for high power laser optics.

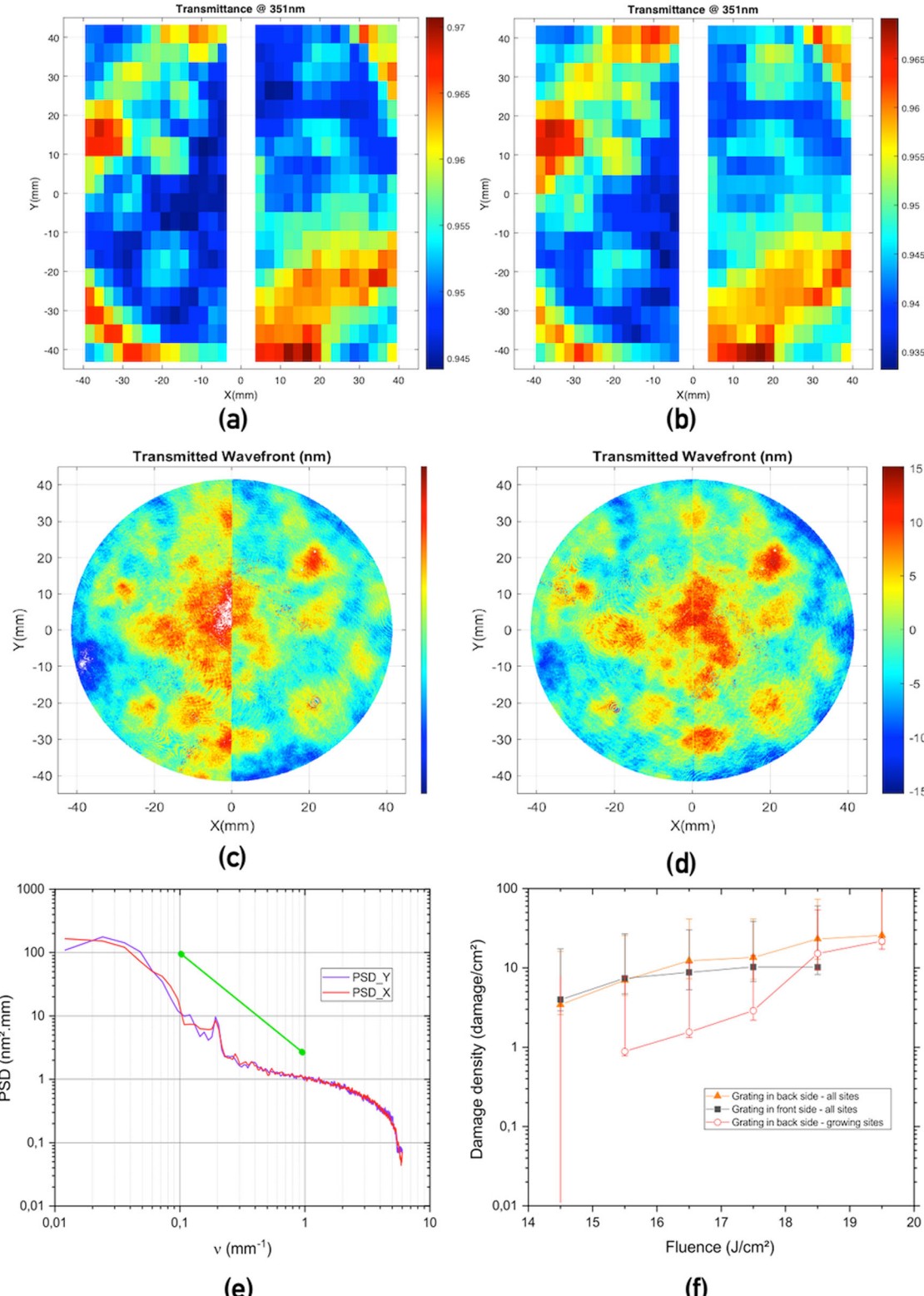

**Fig. 2 | Optical characterization of the full-silica quarter waveplate.**
**a**, **b** Transmission measurements performed at the wavelength of 351 nm, in normal incidence, linear polarization oriented in the vertical direction (**a**) or horizontal direction (**b**). **c**, **d** Transmission wavefront of the quarter waveplate measured at 1064 nm at normal incidence over a 83 mm diameter centered on the clear aperture of the waveplate, **c** polarization oriented in the vertical direction., **d** polarization oriented in the horizontal direction. The white vertical thin line corresponds to the junction between the two areas of the waveplate associated with left and right circular polarization. **e** Power spectral Density (PSD) evaluated from (**d**): the green line presents a typical specification envision for a high power laser. **f** Quarter waveplate damage density. Measurements performed at the wavelength of 355 nm, pulse duration of 7.5 ns with grating placed either in the front or in the back side with respect to the testing beam. Error bars represent intervals of confidence calculated at two standard deviations.

In order to assess the interest of silica waveplates for decreasing the laser filamentation impact on silica windows, we first need to verify their laser resistance to high laser fluences. The laser induced damage resistance was measured in raster scan mode at the wavelength 355 nm using the procedure described in section 3.5. Fluences were re-scaled and expressed at the pulse duration of 3 ns using a $\tau^{0.5}$ relation as established by Stuart in the infrared spectrum[50] and by Grua in the ultraviolet spectrum[51]. Experiments were carried out for two configurations: when the meta-optics is placed on the frontside and the backside with respect to the laser damage testing beam. The raster scan procedure was performed on an area of 4 cm² and the results are presented in Fig. 2f. The first result is that no damage can be observed on the substrate (initiated and therefore growing) below 20 J/cm². When the meta-optics is placed on the backside, damage sites are initiated at 14.5 J/cm² and we measure a damage density of growing sites of 1/cm². When the meta-optics is placed on the frontside, laser damage sites are initiated with a similar density as for the backside configuration, but none of them grows. A similar behavior was observed on transmission gratings used for beam steering and for focusing laser beams of the Megajoule laser[52,53]. This result is assumed to originate from the direction of expansion of the plasma created during the laser/material interaction that shields the damage when the diffractive or meta-optics is in the frontside configuration[54,55].

### Laser filamentation under linear or circular polarization

We now investigate the impact of a linear-to-circular polarization conversion on the laser filamentation by controlling the orientation of the full-silica quarter waveplate characterized in the previous section. For that purpose, filamentation experiments are carried out on a dedicated laser damage testing set-up described in Fig. 3 (see the Methods section "Filamentation experiments") at the wavelength of 351 nm using a 40 mm thick fused silica polished sample.

The sample is bevelled at the corners and polished on the side to facilitate the observation of the filaments. The laser beam is focused on the sample with a long focal lens (3000 mm) so that the beam can be considered as collimated along the sample. At the sample plane, the laser beam features a top-hat profile with a 35% spatial contrast and a circular section of 5 mm in diameter together (Fig. 2b) with a 3 ns flat-in-time pulse temporal shape. The spatial profile and the energy of the laser beam are measured in a plane optically equivalent to the sample plane as depicted in Fig. 2a. The silica waveplate is placed before the focusing lens and its orientation can be controlled by a simple rotation around the laser beam propagation axis. The laser beam propagates through one of the two zones only. Two orientations are of particular interest for this experiment: (i) when the incident linear polarization is aligned with the neutral axis of a zone of the waveplate, the linear polarization is preserved and (ii) when the incident linear polarization is oriented at 45° from the neutral axis, the linear polarization is converted into a circular polarization. Since the two orientations are associated with an excellent transmission efficiency, this simple method allows the waveplate to act either as a neutral component (1) or as a quarter waveplate (2). Let us remark that beam diagnostics of the experimental set-up are disturbed by the modification of the polarization state. However, these deviations are taken into account by recalibrating the diagnostics after each polarization switch. The sample volume is imaged by observing through the polished side with a long-distance working microscope (Fig. 2a). Let us stress that the mean fluence of the laser beam must be distinguished from peak fluences, and that laser filamentations are triggered by the peaks of the fluences. Therefore the side view imaging system is set to image the area at

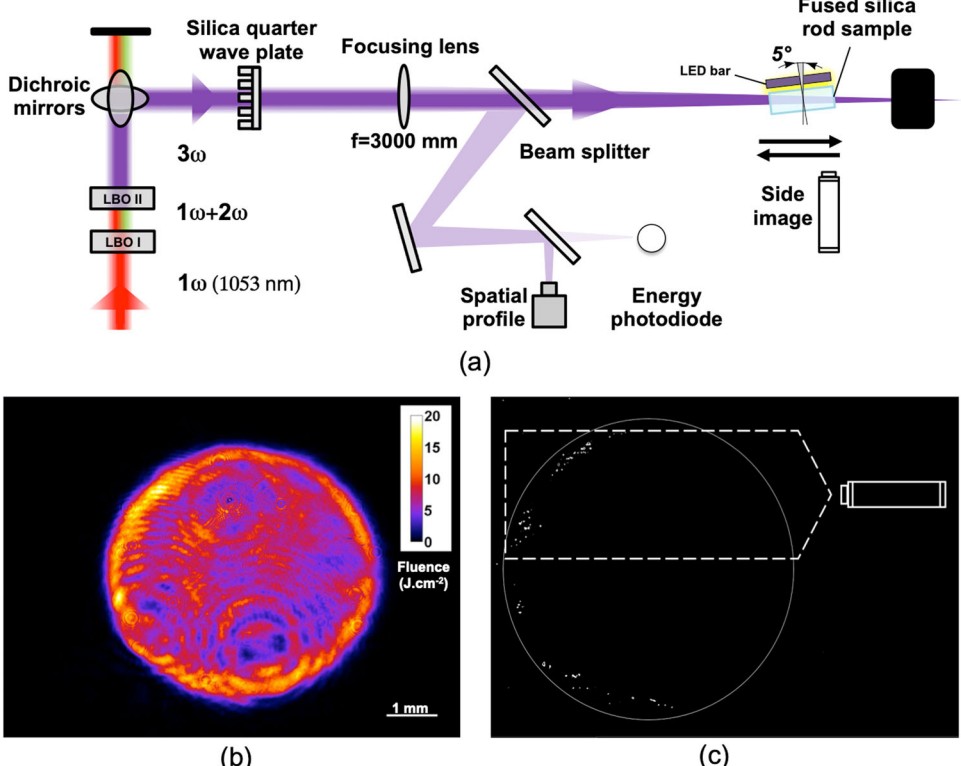

(a)

(b)

(c)

**Fig. 3 | Optical set-up and filamentation imaging. a** Representation of the optical set-up used to perform the filamentation experiments. **b** Beam profile at the wavelength of 351 nm at the entrance of the fused silica thick substrate exhibiting a contrast of 35%. Large laser fluences can be observed on the external zone of the beam. **c** Front view of the thick silica sample featuring filaments exiting the surface. The white circle indicates the contour of the laser beam. Representation of the imaging set-up in dark field configuration for monitoring the filaments at the location of the highest fluence peaks.

which the fluence of the laser beam is the highest and where most filaments are initiated (Fig. 2c). Further image processing combining a Sobel filter, a Hough linear transform and image binarization were performed to ease filaments identification.

Laser filamentation experiments are performed at two different laser fluences, 5.7 and 7.0 J/cm$^2$, and with a linear or a circular polarization state. This corresponds to four different configurations and each configuration is tested twice. Dark field images acquired for these four configurations are shown in Fig. 4. At the laser fluence of 5.7 J/cm$^2$, if linear polarization yields many filaments in the silica sample, a linear-to-circular polarization conversion fully removes the generation of filaments inside the sample. An increase of the laser fluence from 5.7 to 7.0 J/cm$^2$ yields a rise of filaments in linear polarization. A polarization conversion from linear to circular has a strong impact on the filamentation, since it significantly decreases the number of filaments, to a level below that obtained with a linear polarization at 5.7 J/cm$^2$. Dark field images processed by a Sobel filter and a Hough transform are therefore analyzed in order to extract the number of pixels corresponding to a filament. Let us stress that the experiment is carried out in 3D while the imaging system reduces the dimension to a 2D image. The number of pixels extracted from the images for the four configurations (Fig. 4e) must therefore be considered as a qualitative result. The case corresponding to a circular polarization at 5.7 J/cm$^2$ is set equal to 0 since no filament is observed in the image. For the three other configurations, the number of pixels corresponding to filaments is plotted as a function of the distance from the chamfer. The principal result is that a circular polarization at 7.0 J/cm$^2$ (green line in Fig. 4e) allows us to decrease the number of pixels associated with filaments to a level lower than that obtained with a linear polarization at 5.7 J/cm$^2$.

Full-silica meta-optics offer a very promising approach to tailor the polarization states of high energy laser beams in the UV spectrum (351 nm). The most advanced techniques to fabricate wave plates are GLAD (Glancing Angle Deposition)[56] and liquid crystals[57]. GLAD is very interesting for the development of quarter waveplates since it exhibits a high resistance to laser damage. Its main limitations are due to the roughness of the multiple interfaces that can induce losses, and to the scalability of the process to large surfaces. Liquid crystals have also been developed to fabricate various wave plates. In this latter case, the main limitation is linked with the laser induced damage resistance in the UV. Full-silica meta-optics provide a very promising approach for tailoring the polarization states in the final stage of high-energy laser facilities since (i) they exhibit very high laser induced damage thresholds together with excellent optical properties in terms of transmission efficiency and phase; (ii) the top-down lithography offers a high degree of control on the nanostructuring; (iii) the orientation of the ordinary axis of the meta-waveplate can be tuned and different zones and orientations can be associated on a single meta-optics. The main challenge is to upscale this technology to half-meter scale in order to allow its implementation in current high-energy laser facilities targeting inertial fusion ignition. Bringing nanostructured meta-optics to sub-meter scale is one of the most important challenges in nanotechnologies that is currently benefiting from the interest of well-established companies in microelectronics.

In summary, this study shows how the polarization state of high energy laser beams at 351 nm can be easily controlled by rotating a 110 × 110 mm$^2$ full-silica planar meta-optics exhibiting high transmission efficiency, higher than 98%, and high resistance to laser damage with no growing damage sites measured at 14.5 J/cm$^2$. We demonstrate the possibility to control the orientation of the ordinary axis of the waveplate at ± 45° and to associate two orientations on the same surface, offering the possibility to obtain a left or right circular polarization with the same optical component. The strong influence of the polarization state on laser filamentation is evidenced by estimating the filamentation rate of the sample for four different configurations in terms of laser fluence and polarization state. We report a level of

filamentation obtained in a circular polarization lower than that obtained with a linear polarization for a laser fluence 20% smaller.

## Methods

### Nanofabrication
Manufacturing of the meta-optics is performed using a CMOS process on 200 mm fused silica wafers of thickness 0.725 mm. Samples are first coated with a 50 nm Ti hard mask and a photo-resist layer. The photo-resist layer is patterned using an electron beam lithographic device (Variable Shaped Beam SB3054Model from Vistec) on the full 110 × 110 mm$^2$ clear aperture. The continuity between the two patterned waveplate areas is achieved at this step. The pattern is then transferred into the fused silica using a reactive ion etcher. Residues of photoresists and hard mask are then removed using classical CMOS cleaning technics to end-up with a bare fused silica nanostructure. A full description of the manufacturing process flow is available in ref. 46. Two identical fused silica quarterwave plates were manufactured: the first one is dedicated to optical characterization experiments on the wavefront quality, transmission efficiency and filamentation experiments while the second one is dedicated to laser damage resistance measurements.

### Photometry
The transmission efficiency of the waveplate is measured using a small beam single point scanning photometer capable of handling large size optical components up to 400 × 800 mm$^{258}$. The laser scanning beam size is 8 mm at 98% of encircled energy. The system is able to perform measurements in the two *P* and *S* polarizations, varying incidence, and at the three wavelengths of 351, 532 and 1053 nm thanks to a Q-switched Nd:YLF pulsed laser. Measurements were performed at normal incidence at the wavelength of 351 nm with a scanning sampling step of 2 mm other the clear aperture of the samples. Under these conditions, the typical total measurement uncertainty is estimated to be 0.0042 with a confidence level of more than 95%[59].

### Linear retardance
Linear retardance was measured at the Laboratory for Laser Energetics (Rochester, USA) using a Mueller polarimeter from Hind Instrument Inc. (Exicor 450XT). The system operates in ambient air at the wavelength of 355 nm and is able to perform scanning over a large area of up to 450 mm in diameter to estimate point by point the relevant elements of the Mueller matrix. Typical errors of <0.1% for linear retardance and 0.1° for angles can be obtained with this system[60]. Scanning was performed with a sampling step of 0.5 mm on the 110 × 110 mm$^2$ clear aperture of the quarter waveplate, and linear retardance and fast axis orientation from horizontal axis were only collected.

### Transmitted wavefront
The transmitted wavefront was measured using a Zygo GPI 4" interferometer equipped with a 1024 × 1024 pixels camera and operating at the wavelength of 1064 nm in linear polarization[61]. The wavefront measurement is carried out with an aperture of 83 mm centered on the clear aperture of the waveplate; polarization is oriented in a vertical or horizontal direction, i.e., along the *Oy* or *Ox* axis respectively as depicted in Fig. 1b. Data were acquired with Metropro software from Zygo and post-processed using ANAPHASE software. ANAPHASE aims to perform the various mathematical operations to compare measurements to LMJ optical specifications in terms of power, astigmatism, RMS and Power Spectral Density in the spatial period bands of interest for high-power laser optics[49]. Here, we brought a particular attention to mid-spatial frequencies in the [1–10 mm] period range, which are of major importance for large inertial confinement fusion optics[15,48]. Previous works using

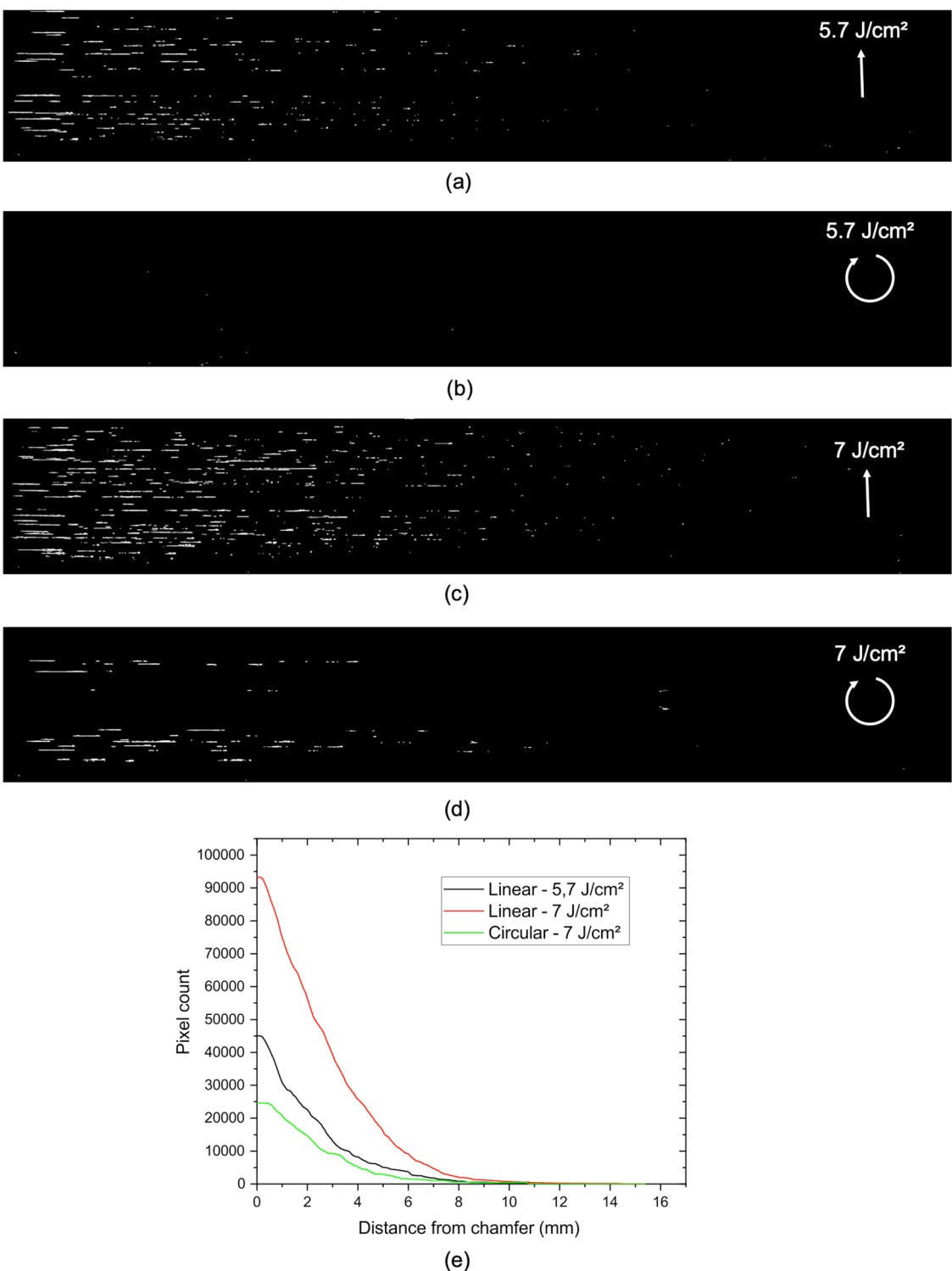

**Fig. 4 | Impact of the linear-to-circular polarization conversion on the laser filamentation.** Dark field images of the fused silica sample after laser damage filamentation experiments carried out at different laser fluences and polarization states. The laser beam comes from the right side. It is characterized by a pulse duration of 3 ns, flat in time, and a wavelength of 351 nm. The left edge of the images corresponds to the start of the bevel sample, located 1 mm away from the sample exit face. Images correspond to a surface of the sample of 2.5 × 15 mm². Images show the sample after the filamentation experiment performed at a mean fluence of 5.7 J/cm² (**a**, **c**) and laser (**b**, **d**), with a linear (**a**, **c**) and circular polarization (**b**, **d**). **e** Number of white pixels in (**a**, **c**, **d**) as a function of the distance from the chamfer (no pixel can be counted in (**b**)). Green line: circular polarization at 7.0 J/cm²; black line: linear polarization at 5.7 J/cm²; red line: linear polarization at 7.0 J/cm².

calibrated phase plates and detailed elsewhere were carried out to characterize the optical response of the interferometer in terms of RMS and power spectral density in the [1–10 mm] period domain[61]. In addition, one dimensional mean PSD along $X$ xand y axis are computed from the transmitted wavefront maps. These PSD are compared to a decreasing one dimensional PSD in the form of $Av^{-2.5}$ ($A = 3$) in the [0.1–1 mm$^{-1}$] range to mimic the typical PSD required for high power laser optics[48,49].

## Laser damage resistance

The laser damage resistance was estimated on a set-up dedicated to large scale optics. The laser characteristics are close to those featured by the LMJ facility: wavelength of 355 nm and pulse duration of 7.5 ns. The frequency of a nanosecond Nd:YAG laser emitting at 1064 nm is tripled to get a wavelength of 355 nm. The laser beam is then focused to irradiate the surface of the sample. The laser delivers ~600 mJ at a nominal repetition rate of 10 Hz. The laser beam is $P$-polarized and focused into the sample by a convex lens which focal length is $f = 8000$ mm. It induces a depth of focus (DOF) higher than the thickness of the metamaterial waveplate. By this way, the beam shape is constant along the DOF meaning that the shape and the diameter of the laser beam were the same on the front and on the rear surfaces of the wave plate. The beam features a spatial Gaussian distribution at the focal plane with a diameter of the order of 700 μm at 1/e. The intensity profiles of the laser pulses were controlled by means of a phase-modulated injection seeder that allows for operating the nanosecond Nd:YAG Q-switched laser with pulses having both a large spectral bandwidth and a smooth temporal waveform. This smooth temporal waveform reduces the impact of the Kerr effect and thanks to the large spectral bandwidth suppresses also the stimulated Brillouin scattering[62]. Diagnostics allow for a precise determination of shot-to-shot beam spatial profiles, pulse temporal profiles, and beam energy deposited on the tested specimen. Thus, the determination of the absolute fluence is measured with an accuracy of up to ±10%[63]. Damage detection is performed in real time with a He-Ne laser beam probe colinear to the test beam. The size and the morphology of the damage sites are visualized in situ with a long-working distance objective microscope with a resolution of 10 μm. Rasterscan experiments were performed at low fluences to detect rare defects that reduce the lifetime of large optics in high energy laser systems. Each detected damage site was then irradiated with series of 200 shots at a growth fluence of 8 J/cm² to check whether damage sites grew or not. In that way, damage densities of only growing damage sites are also estimated. A site whose diameter is >250 μm and/or whose initial diameter has been multiplied by 10 is considered to be a growing site. Intervals of confidence for damage densities are calculated at two standard deviations. Further details on damage testing set-up, procedure and uncertainty evaluation are available in refs. 58,64,65.

## Filamentation experiments

The filamentation experiments were performed on a dedicated facility called MELBA. This customizable laser damage testing set-up with an aperture of a few millimeters is dedicated to laser damage studies of optical materials. The set-up and its diagnostics are schematized in Fig. 3. The system is based on Nd:phosphate flash pumped rod amplifiers emitting up to 7 J at 1053 nm for a beam diameter of 13 mm and repetition rate of 1 shot per minute. LBO (lithium triborate) crystals are used to frequency convert light at 351 nm; up to 4 J are obtained at this wavelength with a beam diameter of up to 7 mm. Pulse duration can be programmed from 1 ns to 20 ns and an anti-Brillouin phase modulation at 2 GHz is also available to limit Brillouin scattering during thick optics damage experiments. Diagnostics allow precise shot-to-shot energy, temporal and spatial profile measurements. A full description of this optical set-up is available in ref. 66.

## Data availability

The data that support the findings of this study are available in a Zenodo repository under accession code https://zenodo.org/record/8138375.

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

## Acknowledgements

The authors thank B. Hoffman and colleagues from the Laboratory for Laser Energetics, University of Rochester, USA for Mueller matrix

retardance measurements. T. Donval and C. Ameil from CEA Cesta / DLP, for laser damage testing and photometric measurements. G. Hallo et Y. Abdelmoumni Prunes from CEA Cesta / DLP, for filamentation image processing.

## Author contributions

The fabrication process of the experimental samples was developed and performed by P.B. Optical metrology was analyzed and supervised by J.D. Laser damage experiments were performed by S.G., N.R., J.F.G., and were supervised by L.L. Numerical modeling was performed by N.B. J.N. and N.B. brought the idea and designed the experiment. J.N., N.B., S.G., and L.L. wrote the manuscript with contributions of all co-authors.

## Competing interests

The authors declare no competing interest.
