## [Peer Review File · Nature Communications]

Linear-to-Circular Polarization Conversion with Full-Silica Meta-Optics to Reduce Nonlinear Effects in High-Energy LasersREVIEWER COMMENTS

Reviewer #1 (Remarks to the Author):

The article demonstrated that a quarter waveplate based on form birefringence of high period grating structure was employed in high energy laser system to compress the nonlinear self-focusing effect, and to improve the ability of high power laser facility. It is a milestone for application of nanograting silica element in large size. Its nanofabrication, wavefront measurement, transmission measurement, and birefringence measurement are technic challenge. The manuscript is well-organized and well-written. There are some suggestions which are listed below:

1, ".....understanding and limiting FM-to-AM," the full name of the abbreviation should be given When it first appears.

2, "quartz are routinely used in optical systems, they cannot be mounted on high-energy laser chains due to laser damage." Quartz possesses very high laser damage threshold, similar to the fused silica. Properly it never used in large laser beam due to the limit of the size of quartz crystal.

3, In part 2.1, form birefringence of the nanostructure should be given to help reader understand the physics. PLS see: Born M and Wolf E 1999 Principles of Optics (Cambridge: Cambridge University Press)

, , n is about 0.11 as your case for ideal structure.

4, in Fig.4.(c), the image is 2D, but the beam in silica rod is 3D. Those curves can't give a quantitative value. My suggestion is to move the Fig.4(c).

5, what's consideration for division into two orthogonal parts? One orientation nanograting can serve as a quarter waveplate.

Reviewer #2 (Remarks to the Author):

The paper details some experimental validation of a novel meta-optic waveplate for high-energy lasers previously reported by the same authors. The methodology appears sound and is based on previously reported and referenced work. The results and conclusions drawn are of significant interest.

The paper is reasonably well written but requires a good read through for English/Grammar issues as the impact is being lost somewhat.

The paper is missing a discussion of the limitations of the work and what is next. A comparison of the results with other work in this area would help to draw out the significance as well.

We thank both reviewers for their willingness to review our manuscript. We are glad to read their positive appreciations of our work. We have carefully read their reports and addressed all their comments.

We provide below a point-by-point answer to all their comments.

Our comments are in purple and our revisions are in green. We provide a redlined edit of the revised manuscript so that you can easily track the revisions brought to the manuscript.

Reviewer#1. The article demonstrated that a quarter waveplate based on form birefringence of high period grating structure was employed in high energy laser system to compress the nonlinear self-focusing effect, and to improve the ability of high power laser facility. It is a milestone for application of nanograting silica element in large size.

We thank the reviewer for this positive appreciation of our work.

Its nanofabrication, wavefront measurement, transmission measurement, and birefringence measurement are technic challenge. The manuscript is well-organized and well-written.

We are pleased to see the reviewer acknowledging the overcoming of several challenges and the combination of nanofabrication achievements associated with a large set of characterization techniques to provide the results that are reported in this manuscript.

There are some suggestions which are listed below:

We thank the reviewer for pointing out to our attention these pertinent comments which we have addressed.

1, "...understanding and limiting FM-to-AM," the full name of the abbreviation should be given When it first appears.

We added the full name when first given :

"understanding and limiting FM-to-AM (Frequency Modulation-to-Amplitude Modulation)"

2, "quartz are routinely used in optical systems, they cannot be mounted on high-energy laser chains due to laser damage." Quartz possesses very high laser damage threshold, similar to the fused silica. Properly it never used in large laser beam due to the limit of the size of quartz crystal.

We agree with the reviewer that the typical size of the optics in high-energy lasers is not compatible with the crystal nature of quartz. We also agree with the fact that it is the size issue that prevents quartz wave plates to be used in large laser facilities. We modified accordingly this sentence:

"if commercial wave-plates based on birefringent materials such as quartz are routinely used in optical systems, they cannot be implemented in high-energy laser chains

because the crystalline nature of quartz is not compatible with the large size of the optics mounted in such laser facilities”.

3, In part 2.1, form birefringence of the nanostructure should be given to help reader understand the physics. PLS see: Born M and Wolf E 1999 Principles of Optics (Cambridge: Cambridge University Press)

, , D_n is about 0.11 as your case for ideal structure.

We added a description of the birefringence on the revised manuscript:

“This meta-optics behaves as a uniaxial crystal [47] and the difference between the ordinary and extraordinary index Δn can be easily estimated from the linear retardance and the nanostructuring depth at $\Delta n=0.0825$.”

4, in Fig.4.(c), the image is 2D, but the beam in silica rod is 3D. Those curves can't give a quantitative value. My suggestion is to move the Fig.4(c).

We thank the reviewer for pointing out the difference between the 3D beam and the 2D image. This discussion convinced us to provide more details in the main manuscript and to discuss this important point. We think that Fig. 4e provide valuable results but we agree with the reviewer on the fact that it displays qualitative results. This point is now clarified:

“Let us stress that the experiment is carried out in 3D while the imaging system reduces the dimension to a 2D image. The number of pixels extracted from the images to estimate the filamentation rate for the 4 configurations (see Fig.4(e)) must therefore be considered as a qualitative result.”

5, what's consideration for division into two orthogonal parts? One orientation nanograting can serve as a quarter waveplate.

We fully agree with the reviewer on the fact that one orientation is enough to design a quarter wave plate. We believe that in future developments, the control of the orientation and the association of different orientations on the same metasurface could be important, in particular to yield beams with orthogonal polarizations. Here, our motivation was to evidence the ability of this top down approach to associate on the same metasurface, and without discontinuity, two perpendicular orientations yielding left and right circular polarizations. This point is discussed in the conclusion section:

“We evidence the possibility to control the orientation of the ordinary axis of the waveplate at $\pm 45^\circ$ and to associate two orientations on the same surface, offering the possibility to yield a left or right circular polarization with the same optical component.”

Reviewer#2. The paper details some experimental validation of a novel meta-optic waveplate for high-energy lasers previously reported by the same authors. The methodology appears sound and is based on previously reported and referenced work. The results and conclusions drawn are of significant interest.

We thank the reviewer for this positive appreciation of our manuscript.

The paper is reasonably well written but requires a good read through for English/Grammar issues as the impact is being lost somewhat.

We thoroughly proofread the whole manuscript and improved the overall quality of grammar and English.

The paper is missing a discussion of the limitations of the work and what is next. A comparison of the results with other work in this area would help to draw out the significance as well.

We agree with the reviewer that an overall discussion on the strengths and limitations of this method was missing. To address this point, we added a discussion before the conclusion on the strength and weakness of this approach and a comparison with other technologies, in particular liquid crystals and GLAD technologies:

“Full-silica meta-optics offer a novel and very promising approach to tailor the polarization states of high energy laser beams in the UV spectrum (351 nm). The most advanced techniques to fabricate wave plates are GLAD (Glancing Angle Deposition) [55] and liquid crystals [56]. GLAD is very interesting for the development of quarter waveplates with high resistance to laser damage. Its main limitations are due to the roughness of the multiple interfaces that can induce losses, and to the scalability of the process to large surfaces. Liquid crystals have also been developed to fabricate various wave plates. In this latter case, the main limitation is linked with the laser induced damage resistance in the UV. Providing a novel route based on full-silica meta-optics for tailoring the polarization states in the final stage of high-energy laser facilities is of strong interest since (i) they exhibit very high laser induced damage thresholds together with excellent optical properties in terms of transmission efficiency and phase; (ii) the top-down lithography offers a high degree of control on the nanostructuring; (iii) the orientation of the ordinary axis of the meta-waveplate can be tuned and different zones and orientations can be associated on a single meta-optics. The main challenge is to upscale this technology to half-meter scale in order to allow its implementation in current high-energy laser facilities targeting inertial fusion ignition. Bringing nanostructured meta-optics to sub-meter scale is one of the most important challenges in nanotechnologies that is currently benefiting from the interest of well-established companies in microelectronics.”

REVIEWERS' COMMENTS

Reviewer #1 (Remarks to the Author):

My questions have been addressed well in the new version. Especially the $\pm 45^\circ$ orientated meta-surface demonstrate ability of phase delay with complex structure in large size. I recommend the manuscript for publication.

Reviewer #2 (Remarks to the Author):

The paper is much improved and has taken account of the comments and recommendations that reviewers have made.

I would now recommend publication as is.